# INTROSPECTION: ACCELERATING NEURAL NETWORK TRAINING BY LEARNING WEIGHT EVOLUTION

**Abhishek Sinha**[*]
Department of Electronics and Electrical Comm. Engg.
IIT Kharagpur
West Bengal, India
abhishek.sinha94 at gmail dot com

**Mausoom Sarkar**
Adobe Systems Inc, Noida
Uttar Pradesh, India
msarkar at adobe dot com

**Aahitagni Mukherjee**[*]
Department of Computer Science
IIT Kanpur
Uttar Pradesh, India
ahitagnimukherjeeam at gmail dot com

**Balaji Krishnamurthy**
Adobe Systems Inc, Noida
Uttar Pradesh, India
kbalaji at adobe dot com

## ABSTRACT

Neural Networks are function approximators that have achieved state-of-the-art accuracy in numerous machine learning tasks. In spite of their great success in terms of accuracy, their large training time makes it difficult to use them for various tasks. In this paper, we explore the idea of learning weight evolution pattern from a simple network for accelerating training of novel neural networks.

We use a neural network to learn the training pattern from MNIST classification and utilize it to accelerate training of neural networks used for CIFAR-10 and ImageNet classification. Our method has a low memory footprint and is computationally efficient. This method can also be used with other optimizers to give faster convergence. The results indicate a general trend in the weight evolution during training of neural networks.

## 1 INTRODUCTION

Deep neural networks have been very successful in modeling high-level abstractions in data. However, training a deep neural network for any AI task is a time-consuming process. This is because a large number of parameters need to be learnt using training examples. Most of the deeper networks can take days to get trained even on GPU thus making it a major bottleneck in the large-scale application of deep networks. Reduction of training time through an efficient optimizer is essential for fast design and testing of deep neural nets.

In the context of neural networks, an optimization algorithm iteratively updates the parameters (weights) of a network based on a batch of training examples, to minimize an objective function. The most widely used optimization algorithm is Stochastic Gradient Descent. Even with the advent of newer and faster optimization algorithms like Adagrad, Adadelta, RMSProp and Adam there is still a need for achieving faster convergence.

In this work we apply neural network to predict weights of other in-training neural networks to accelerate their convergence. Our method has a very low memory footprint and is computationally efficient. Another aspect of this method is that we can update the weights of all the layers in parallel.

---

[*]This work was done as part of an internship at Adobe Systems, Noida

## 2 RELATED WORK

Several extensions of Stochastic Gradient Descent have been proposed for faster training of neural networks. Some of them are Momentum (Rumelhart et al., 1986), AdaGrad (Duchy et al., 2011), AdaDelta (Zeiler, 2012), RMSProp (Hinton et al., 2012) and Adam (Kingma & Ba, 2014). All of them reduce the convergence time by suitably altering the learning rate during training. Our method can be used along with any of the above-mentioned methods to further improve convergence time.

In the above approaches, the weight update is always a product of the gradient and the modified/unmodified learning rate. More recent approaches (Andrychowicz et al., 2016) have tried to learn the function that takes as input the gradient and outputs the appropriate weight update. This exhibited a faster convergence compared to a simpler multiplication operation between the learning rate and gradient. Our approach is different from this, because our forecasting Network does not use the current gradient for weight update, but rather uses the weight history to predict its future value many time steps ahead where network would exhibit better convergence. Our approach generalizes better between different architectures and datasets without additional retraining. Further our approach has far lesser memory footprint as compared to (Andrychowicz et al., 2016). Also our approach need not be involved at every weight update and hence can be invoked asynchronously which makes it computationally efficient.

Another recent approach, called Q-gradient descent (Fu et al., 2016), uses a reinforcement learning framework to tune the hyperparameters of the optimization algorithm as the training progresses. The Deep-Q Network used for tuning the hyperparameters itself needs to be trained with data from any specific network $N$ to be able to optimize the training of $N$. Our approach is different because we use a pre-trained forecasting Network that can optimize any network $N$ without training itself by data from $N$.

Finally the recent approach by (Jaderberg et al., 2016) to predict synthetic gradients is similar to our work, in the sense that the weights are updates independently, but it still relies on an estimation of the gradient, while our update method does not.

Our method is distinct from all the above approaches because it uses information obtained from the training process of existing neural nets to accelerate the training of novel neural nets.

## 3 PATTERNS IN WEIGHT EVOLUTION

The evolution of weights of neural networks being trained on different classification tasks such as on MNIST and CIFAR-10 datasets and over different network architectures (weights from different layers of fully connected as well as convolutional architectures) as well as different optimization rules were analyzed. It was observed that the evolution followed a general trend independent of the task the model was performing or the layer to which the parameters belonged to. A major proportion of the weights did not undergo any significant change. Two metrics were used to quantify weight changes:

- *Difference between the final and initial values of a weight scalar:* This is a measure of how much a weight scalar has deviated from its initial value after training. In figure 4 we show the frequency histogram plot of the weight changes in a convolutional network trained for MNIST image classification task, which indicates that most of the weight values do not undergo a significant change in magnitude. Similar plots for a fully connected network trained on MNIST dataset ( figure 6 ) and a convolutional network trained on CIFAR-10 dataset (figure 8 ) present similar observations.

- *Square root of 2nd moment of the values a weight scalar takes during training:* Through this measure we wish to quantify the oscillation of weight values. This moment has been taken about the initial value of the weight. In figure 5, we show the frequency histogram plot of the second moment of weight changes in a convolutional network trained for the MNIST digit classification task, which indicates that most of the weight values do not undergo a significant oscillations in value during the training. Similar plots for a fully

connected network trained on MNIST (figure 7 ) and a convolutional network trained on CIFAR-10 ( figure 9) dataset present similar observations.

A very small subset of the all the weights undergo massive changes compared to the rest.

The few that did change significantly were observed to be following a predictable trend, where they would keep on increasing or decreasing with the progress of training in a predictable fashion. In figures 1, 2 and 3 we show the evolution history of a few weights randomly sampled from the weight change histogram bins of figures 4,6 and 8 respectively, which illustrates our observation.

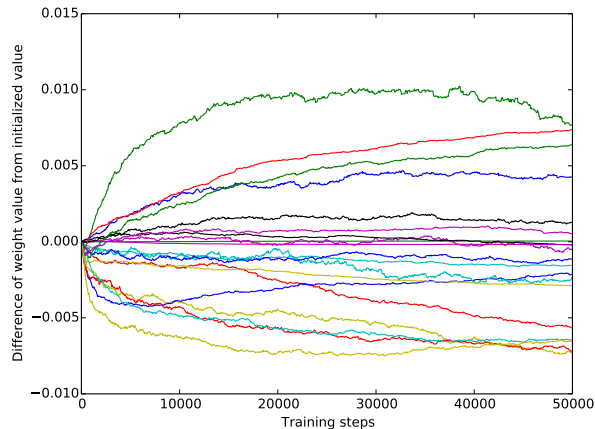

Figure 1: Deviation of weight values from initialized values as a convolutional network gets trained on MNIST dataset using SGD optimizer.

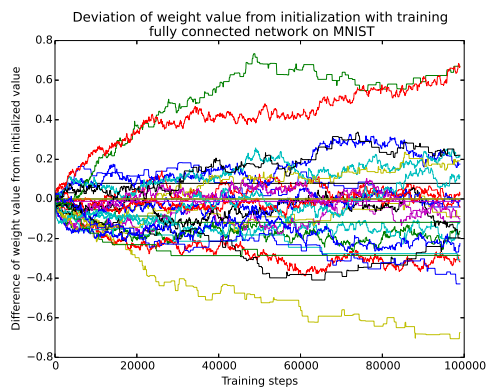

Figure 2: Deviation of weight values from initialized values as a fully-connected network gets trained on MNIST dataset using Adam optimizer..

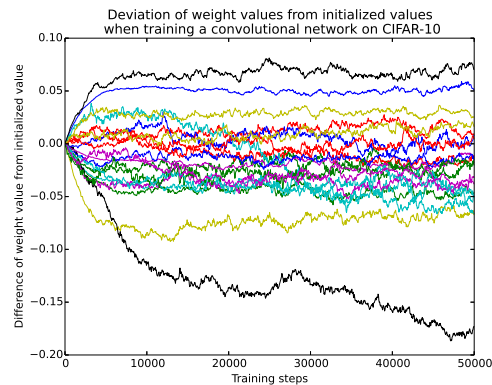

Figure 3: Deviation of weight values from initialized values as CNN gets trained on CIFAR-10 dataset using SGD optimizer.

## 3.1 WEIGHT PREDICTION

We collect the weight evolution trends of a network that is being trained and use the collected data to train a neural network $I$ to forecast the future values of each weight based on its values in the previous time steps. The trained network $I$ is then used to predict the weight values of an unseen network $N$ during its training which move $N$ to a state that enables a faster convergence. The time taken for the forecast is significantly smaller compared to the time a standard optimizer (e.g. SGD) would have taken to achieve the same accuracy. This leads to a reduction in the total training

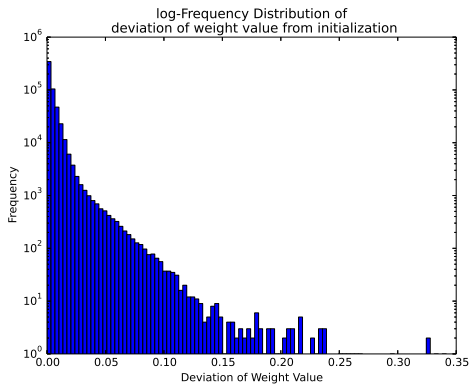

Figure 4: log-Frequency distribution of difference between weight values before and after training for a network $N_0$ trained on MNIST dataset using SGD optimizer.

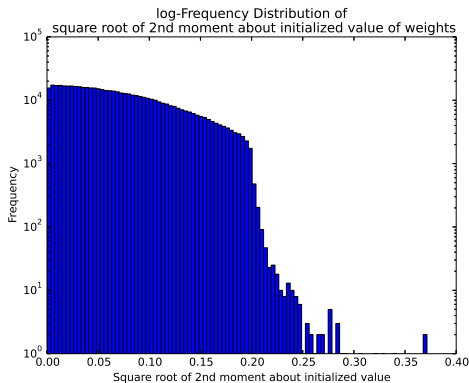

Figure 5: log-Frequency distribution of square root of 2nd moment of a weight value(about initial value) along its training history. The weight values are taken from a network $N_0$ trained on MNIST dataset using SGD optimizer.

time. The predictor $I$ that is used for forecasting weights is a comparatively smaller neural network, whose inference time is negligible compared to the training time of the network that needs to be trained(N). We call this predictor $I$ Introspection network because it looks at the weight evolution during training.

The forecasting network I is a simple 1-layered feedforward neuralnet. The input layer consists of four neurons that take four samples from the training history of a weight. The hidden layer consists of 40 neurons, fully connected to the input layer, with ReLU activation. The output layer is a single neuron that outputs the predicted future value of the weight. In our experiments four was minimum numbers of samples for which the training of Introspection Network $I$ converged.

The figure 10 below shows a comparison of the weight evolution for a single scalar weight value with and without using the introspection network $I$. The vertical green bars indicate the points at which the introspection network was used to predict the future values. Post prediction, the network continues to get trained normally by SGD, until the introspection network $I$ is used once again to jump to a new weight value.

## 4 EXPERIMENTS

### 4.1 TRAINING OF INTROSPECTION NETWORK

The introspection network $I$ is trained on the training history of the weights of a network $N_0$ which was trained on MNIST dataset.The network $N_0$ consisted of 3 convolutional layers and two fully connected layers, with ReLU activation and deploying Adam optimiser. Max pooling(2X2 pool size and a 2X2 stride) was applied after the conv layers along with dropout applied after the first fc layer. The shapes of the conv layer filters were $[5, 5, 1, 8]$, $[5, 5, 8, 16]$ and $[5, 5, 16, 32]$ respectively whereas of the fc layer weight were $[512, 1024]$ and $[1024, 10]$ respectively.The network $N_0$ was trained with a learning rate of $1e - 4$ and batch size of $50$. The training set of $I$ is prepared as follows. A random training step $t$ is selected for each weight of $N_0$ selected as a training sample and the following 4 values are given as inputs for training $I$:

1. value of the weight at step $t$
2. value of the weight at step $7t/10$
3. value of the weight at step $4t/10$
4. at step 0 (i.e. the initialized value)

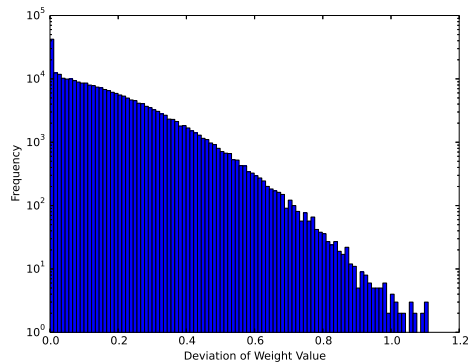

Figure 6: log-Frequency distribution of difference between weight values before and after training for a fully-connected network trained on MNIST dataset using Adam optimizer.

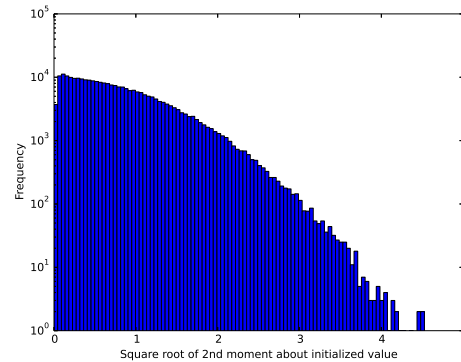

Figure 7: log-Frequency distribution of square root of 2nd moment of a weight value(about initial value) along its training history. The weight values are taken from a fully-connected network trained on MNIST dataset using Adam Optimizer.

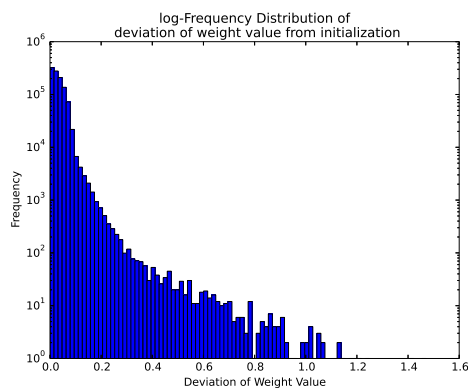

Figure 8: log-Frequency distribution of difference between weight values before and after training for a CNN trained on CIFAR-10 dataset using SGD optimizer.

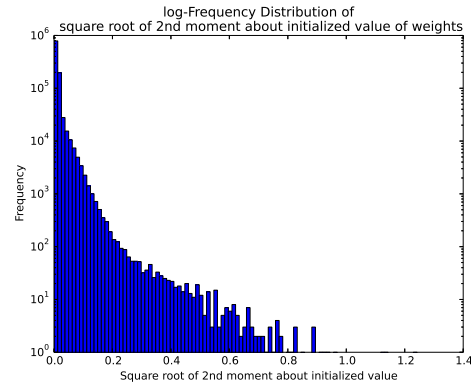

Figure 9: log-Frequency distribution of square root of 2nd moment of a weight value(about initial value) along its training history. The weight values are taken from a CNN trained on CIFAR-10 dataset using SGD Optimizer.

Since a large proportion of weights remain nearly constant throughout the training, a preprocessing step is done before getting the training data for I. The large number of weight histories collected are sorted in decreasing order on the basis of their variations in values from time step 0 to time step t. We choose $50\%$ of the training data from the top 50th percentile of the sorted weights, $25\%$ from the next 25th percentile(between 50 to 75th percentile of the sorted weights) and the remaining $25\%$ from the rest (75th to 100th percentile). Approximately $0.8$ million examples of weight history are used to train $I$. As the weight values are very small fractions they are further multiplied by 1000 before being input to the network I. The expected output of $I$, which is used for training $I$ using backpropagation, is a single scalar  the value of the same weight at step $2t$. This is an empirical choice. For example, any step $kt$ with $k > 1$ can be chosen instead of $2t$. In our experiments with varying the value of $k$, we found that the value of $k = 2.2$ reached a slightly better validation accuracy than $k = 2.0$ on MNIST dataset (see figure 15 ) but, on the whole the value of $k = 2.0$ was a lot more consistent in its out-performance at various points in its history. All the results reported here are with respect to the $I$ trained to predict weight values at $2t$.

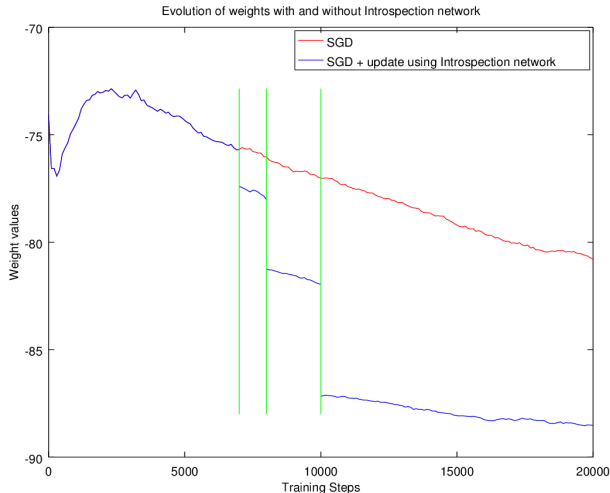

Figure 10: Example of weight update using Introspection Network.

Adam optimizer was used for the training of the introspection network with a mini-batch size of 20.The training was carried out for 30k steps. The learning rate used was 5e-4 which decreased gradually after every 8k training steps. L1- error was used as the loss function for training . We experimented with both L2 error and percentage error but found that L1 error gave the best result over the validation set. The final training loss obtained was 3.1 and the validation loss of the final trained model was 3.4. These correspond to average L1 weight prediction error of 0.0031 and 0.0034 in the training and validation set respectively as the weight values are multiplied by 1000 before they are input to $I$.

## 4.2 Using pre-trained Introspection Network to train unseen networks

The introspection network once trained can be then used to guide the training of other networks. We illustrate our method by using it to accelerate the training of several deep neural nets with varying architectures on 3 different datasets, namely MNIST, CIFAR-10 and ImageNet. We note that the same introspection network $I$, trained on the weight evolutions of the MNIST network $N_0$ was used in all these different cases.

All the networks have been trained using either Stochastic Gradient Descent, or ADAM and the network $I$ is used at a few intermediate steps to propel the network to a state with higher accuracy.We refer to the time step at which the introspection network $I$ is applied to update all the weights as a "jump point".

The selection of the steps at which $I$ is to be used is dependent on the distribution of the training step $t$ used for training $I$. We show the effect of varying the timing of the initial jump and the time interval between jump points in section 4.2.2. It has been observed that $I$ gives a better increase in accuracy when it is used in later training steps rather than in the earlier ones.

All the networks trained using $I$ required comparatively less time to reach the same accuracy as normal SGD training. Also, when the same network was trained for the same time with and without updates by $I$, the former is observed to have better accuracy. These results show that there is a remarkable similarity in the weight evolution trajectories across network architectures,tasks and datasets.

### 4.2.1 MNIST

Four different neural networks were trained using $I$ on MNIST dataset:

1. A convolutional neural network $MNIST_1$ with 2 convolutional layer and 2 fully connected layers(dropout layer after 1st fc layer is also present)with ReLU acitvations for

classification task on MNIST image dataset.Max pooling(2X2 pool size and a 2X2 stride) was applied after every conv layer. The CNN layer weights were of shape $[5, 5, 1, 8]$ and $[5, 5, 32, 64]$ respectively and the fc layer were of sizes $[3136, 1024]$ and $[1024, 10]$.The weights were initialised from a truncated normal distribution with a mean of 0 and std of 0.01. The network was trained using SGD with a learning rate of $1e-2$ and batch size of 50. It takes approximately 20,000 steps for convergence via SGD optimiser. For $MNIST_1$, $I$ was used to update all weights at training step 3000, 4000, and 5000.

2. A convolutional network $MNIST_2$ with 2 convolutional layer and 2 fully connected layers with ReLU acitvations. Max pooling(2X2 pool size and a 2X2 stride) was applied after every conv layer. The two fc layer were of sizes $[800, 500]$ and $[500, 10]$ whereas the two conv layers were of shape $[5, 5, 1, 20]$ and $[5, 5, 20, 50]$ respectively. The weight initialisations were done via xavier intialisation. The initial learning rate was 0.01 which was decayed via the inv policy with gamma and power being $1e-4$ and 0.75 respectively. Batch size of 64 was used for the training.It takes approximately 10,000 steps for convergence . The network $I$ was used to update weights at training step 2500 and 3000.

3. A fully connected network $MNIST_3$ with 2 hidden layers each consisting of 256 hidden units and having ReLU acitvations. The network was trained using SGD with a learning rate of $5e-3$ and a batch size of 100. The initial weights were drawn out from a normal distribution having mean 0 and std as 1.0. For this network the weight updations were carried out at steps 6000, 8000 and 10000.

4. A RNN $MNIST_4$ used to classify MNIST having a LSTM cell of hidden size of 128 followed by a fc layer of shape $[128, 10]$ for classification. The RNN was trained on Adam optimizer with a learning rate of $5e-4$ and a batch size of 128. The weight updations for this network were done at steps 2000,3000 and 4000. Since the LSTM cell uses sigmoid and tanh activations, the RNN $MNIST_4$ allows us to explore if the introspection network, trained on ReLU can generalize to networks using different activation functions.

A comparison of the validation accuracy with and without updates by $I$ is shown in figures 11, 12 ,13 and 14. The green lines indicate the steps at which the introspection network $I$ is used. For the $MNIST_1$ network with the application of the introspection network $I$ at three points, we found that it took 251 seconds and 20000 SGD steps to reach a validation accuracy of 98.22%. In the same number of SGD steps, normal training was able to reach a validation accuracy of only 97.22%. In the same amount of time (251 seconds), normal training only reached 97.92%. Hence the gain in accuracy with the application of introspection network translates to real gains in training times.

For the MNIST2 network, the figure 12 shows that to reach an accuracy of 99.11%, the number of iterations required by normal SGD was 6000, whereas with the application of the introspection network $I$, the number of iterations needed was only 3500, which represents a significant savings in time and computational effort.

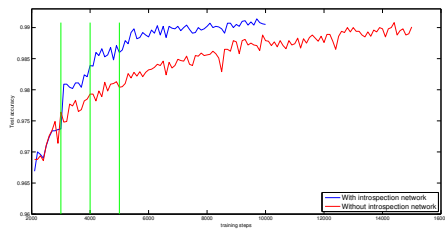

Figure 11: Validation accuracy plot for $MNIST_1$

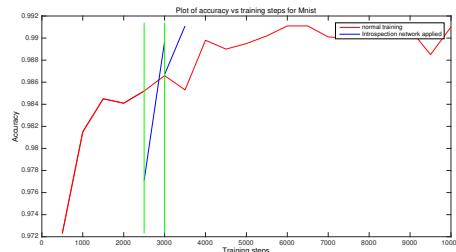

Figure 12: Validation accuracy plot for $MNIST_2$

The initial drop in accuracy seen after a jump in $MNIST_2$ figure 12 can be attributed to the fact that each weight scalar is predicted independently, and the interrelationship between the weight scalars in a layer or across different layers is not taken into consideration. This interrelationship is soon reestablished after few SGD steps. This phenomenon is noticed in the CIFAR and ImageNet cases too.

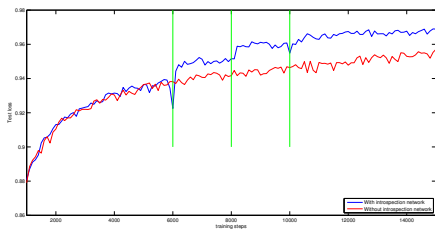 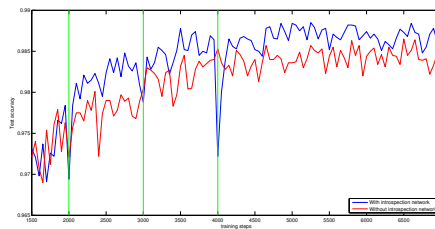

Figure 13: Validation accuracy plot for $MNIST_3$

Figure 14: Validation accuracy plot for $MNIST_4$ Which is an RNN

For $MNIST_3$ after 15000 steps of training,the max accuracy achieved by normal training of network via Adam optimizer was $95.71\%$ whereas with introspection network applied the max accuracy was $96.89\%$. To reach the max accuracy reached by normal training , the modified network(weights updated by $I$) took only 8300 steps.

For $MNIST_4$ after 7000 steps of training, the max accuracy achieved by normal training of network was $98.65\%$ achieved after 6500 steps whereas after modification by $I$ it was $98.85\%$ achieved after 5300 steps. The modified network(weights updated by $I$) reached the max accuracy achieved by normal network after only 4200 steps. It is notable that the introspection network $I$ trained on weight evolutions with ReLU activations was able to help accelerate the convergence of an RNN network which uses sigmoid and tanh activations.

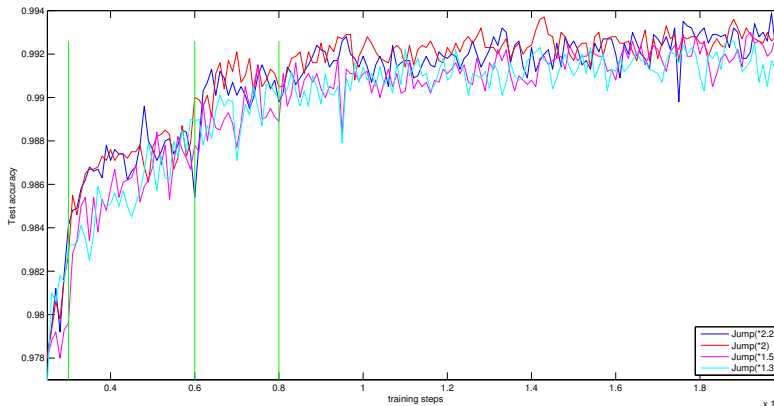

Figure 15: Comparison of introspection networks trained with different jump ratios on $MNIST_1$ network with Adam optimizer.Jump of 2.0 has a more consistent out performance compared to a jump value of 2.2 even though it reaches a slightly higher accuracy

### 4.2.2 CIFAR-10

We applied our introspection network $I$ on a CNN $CIFAR_1$ for classifying images in the CIFAR10 (Krizhevsky, 2009) dataset. It has 2 convolutional layers, 2 fully connected layer and a final softmax layer with ReLU activation function. Max pooling (3X3 pool size and a 2X2 stride) and batch normalization has been applied after each convolutional layer. The two conv layer filter weights were of shape $[5, 5, 3, 64]$ and $[5, 5, 64, 64]$ respectively whereas the two fc layers and final softmax layer were of shape $[2304, 384], [384, 192]$ and $[192, 10]$ respectively. The weights were initialized from a zero mean normal distribution with std of $1e - 4$ for conv layers,0.04 for the two fc layers and $1/192.0$ for the final layer. The initial learning rate used is $0.1$ which is decayed by a factor of $0.1$ after every 350 epochs. Batch size of 128 was used for training of the model which was trained via the SGD optimizer. It takes approximately 40,000 steps for convergence. The experiments on

$CIFAR_1$ were done to investigate two issues. The first was to investigate if the introspection network trained on MNIST weight evolutions is able to generalize to a different network and different dataset. The second was to investigate the effect of varying the timing of the initial jump, the interval between successive jumps and the number of jumps. To investigate these issues, four separate training instances were performed with 4 different set of jump points:

1. $Set_1$ : Weight updates were carried out at training steps 12000 and 17000.
2. $Set_2$ : Weight updates at steps 15000 and 18000 .
3. $Set_3$ : Weight updates at steps 12000 , 15000 and 19000 .
4. $Set_4$ : Weight updates at steps 14000 , 17000 and 20000 .

We observed that for the $CIFAR_1$ network that in order to reach a validation accuracy of 85.7%, we need 40,000 iterations with normal SGD without any intervention with the introspection network $I$. In all the four sets where the introspection network was used, the target accuracy of 85.7% was reached in approximately 28,000 steps. This shows that the introspection network is able to successfully generalize to a new dataset and new architecture and show significant gains in training time.

On $CIFAR_1$, the time taken by $I$ for prediction is negligible compared to the time required for SGD. So the training times in the above cases on $CIFAR_1$ can be assumed to be proportional to the number of SGD steps required.

A comparison of the validation accuracy with and without updates by $I$ at the four different sets of jump points are shown in figures 16, 17, 18 and 19. The results show that the while choice of jump points have some effect on the final result, the effects are not very huge. In general, we notice that better accuracy is reached when the jumps take place in later training steps.

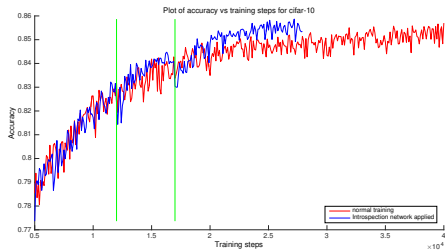

Figure 16: Validation accuracy plot for $CIFAR_1$ with jumps at $Set_1$

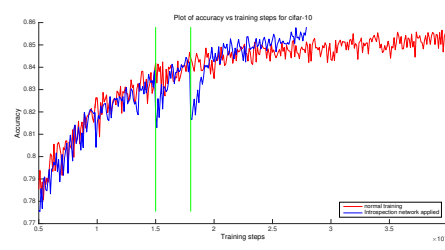

Figure 17: Validation accuracy plot for $CIFAR_1$ with jumps at $Set_2$

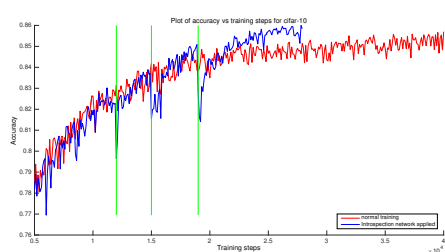

Figure 18: Validation accuracy plot for $CIFAR_1$ with jumps at $Set_3$

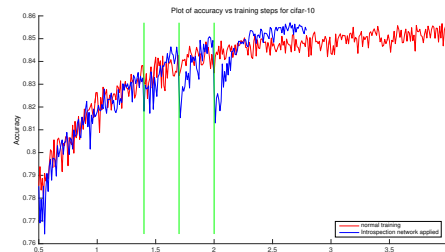

Figure 19: Validation accuracy plot for $CIFAR_1$ with jumps at $Set_4$

### 4.2.3 IMAGENET

To investigate the practical feasibility and generalization ability of our introspection network, we applied it in training AlexNet(Krizhevsky et al., 2012) ($AlexNet_1$) on the ImageNet (Russakovsky

et al., 2015) dataset. It has 5 conv layers and 3 fully connected layers . Max pooling and local response normalization have been used after the two starting conv layers and the pooling layer is there after the fifth conv layer as well. We use SGD with momentum of 0.9 to train this network, starting from a learning rate of $0.01$. The learning rate was decreased by one tenth every $100,000$ iterations. The mini-batch size was 128. It takes approximately 300,000 steps for convergence. The weight updates were carried out at training steps $120,000$ , $130,000$ , $144,000$ and $160,000$ .

We find that in order to achieve a top-5 accuracy of 72%, the number of iterations required in the normal case was 196,000. When the introspection network was used, number of iterations required to reach the same accuracy was 179,000. Again the time taken by $I$ for prediction is negligible compared to the time required for SGD. A comparison of the validation accuracy with and without updates by $I$ is shown in figure 20. The green lines indicate the steps at which the introspection network $I$ is used. The corresponding plot of loss function against training steps has been shown in figure 21.

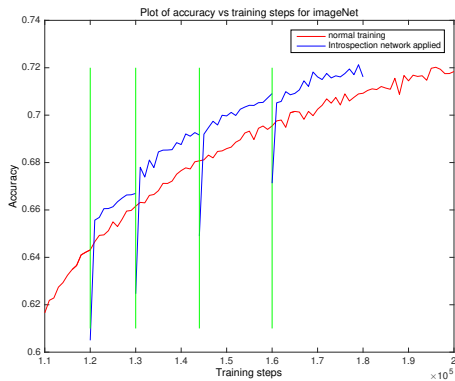

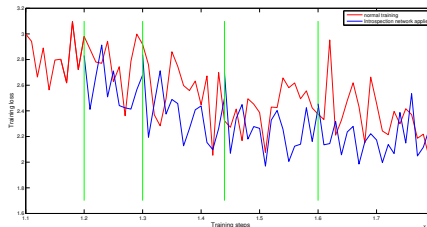

Figure 21: Plot of loss function vs training steps for $AlexNet_1$ on ImageNet

Figure 20: Validation accuracy plot for $AlexNet_1$ on ImageNet

The results on $Alexnet_1$ show that our approach has a small memory footprint and computationally efficient to be able to scale to training practical large scale networks.

## 4.3 COMPARISON WITH BASELINE TECHNIQUES

In this section we provide a comparison with other optimizers and simple heuristics which can be used to update the weights at different training steps instead of updations by introspection network.

## 4.4 COMPARISON WITH ADAM OPTIMIZER

We applied the introspection network on $MNIST_1$ and $MNIST_3$ networks being trained with Adam optimizer with learning rates of $1e-4$ and $1e-3$. The results in figure 22 and figure 23 show that while Adam outperforms normal SGD and SGD with introspection, we were able to successfully apply the introspection network on Adam optimizer and accelerate it.

For $MNIST_1$ the max accuracy achieved by Adam with introspection was $99.34\%$, by normal Adam was $99.3\%$, by SGD with introspection was $99.21\%$ and by normal SGD was $99.08\%$ . With introspection applied on Adam the model reaches the max accuracy as achieved by normal Adam after only 7200 steps whereas the normal training required 10000 steps.

For $MNIST_3$ the max accuracy achieved by Adam with introspection was $96.9\%$, by normal Adam was $95.7\%$, by SGD with introspection was $94.47\%$ and by normal SGD was $93.39\%$ . With introspection applied on Adam the model reaches the max accuracy as achieved by normal Adam after only 8800 steps whereas the normal training required 15000 steps.

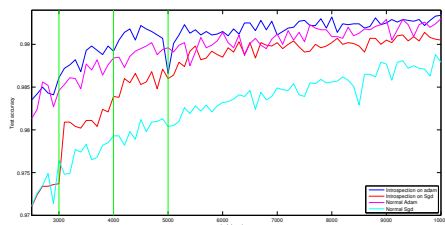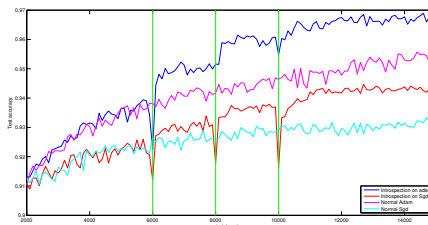

| | |
|---|---|
| Figure 22: Test accuracy comparison for $MNIST_1$ for SGD and Adam optimiser in the presence and absence of introspection. | Figure 23: Test accuracy comparison for $MNIST_3$ for SGD and Adam optimiser in the presence and absence of introspection. |

#### 4.4.1 FITTING QUADRATIC CURVE

A separate quadratic curve was fit to each of the weight values of the model on the basis of the 4 past weight values chosen from history.The weight values chosen from history were at the same steps as they were for updations by $I$. The new updated weight would be the value of the quadratic curve at some future time step.For $MNIST_1$, experiments were performed by updating the weights to the value predicted by the quadratic function at a future timestep which was one of 1.25,1.3 or 1.4 times the current time step. For other higher jump ratios the updates would cause the model to diverge, and lower jump ratios did not show much improvement in performance. The plot showing the comparison in validation accuracy have been shown below in figure 24.

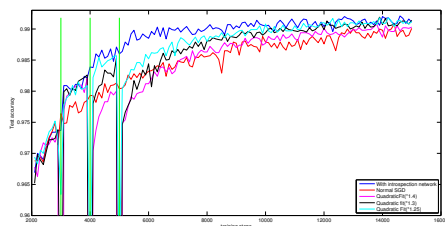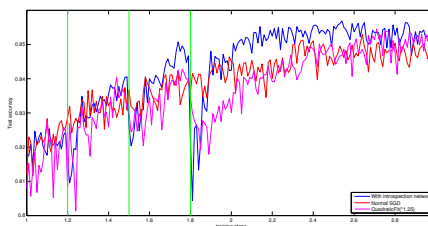

| | |
|---|---|
| Figure 24: Comparison of test accuracy for $MNIST_1$ with weight updations by Introspection and quadratic fit. | Figure 25: Comparison of test accuracy for CIFAR-10 with weight updations by Introspection and quadratic fit. |

The max accuracy achieved with introspection applied was 99.21% whereas with quadratic fit it was 99.19%. We note that even though the best performing quadratic fit eventually almost reaches the same max accuracy than that achieved with introspection network, it required considerable experimentation to find the right jump ratio.A unique observation for the quadratic fit baseline was that it would take the accuracy down dramatically, upto 9.8%, from which the training often never recovers. Sometimes,the optimizers (SGD or Adam) would recover the accuracy, as seen in figure 24. Moreover, the quadratic fit baseline was not able to generalize to other datasets and tasks. The best performing jump ratio of 1.25 was not able to outperform Introspection on the CIFAR-10 dataset, as seen in figure 25.

In the CIFAR-10 case, The maximum accuracy achieved via updations by introspection was 85.6 which was achieved after 25500 steps, whereas with updations by quadratic fit, the max accuracy of 85.45 was achieved after 27200 steps.

For the normal training via SGD without any updations after 30000 steps of training, the max accuracy of 85.29 was achieved after 26500 steps, whereas the same accuracy was achieved by introspection after only 21200 steps and after 27000 steps via updation by quadratic.

### 4.4.2 FITTING LINEAR CURVE

Instead of fitting a quadratic curve to each of the weights we tried fitting a linear curve. Experiments were performed on $MNIST_1$ for jump ratios of $1.1$ and $1.075$ as the higher ratios would cause the model to diverge after 2 or 3 jumps.The result has been shown below in figure 26.

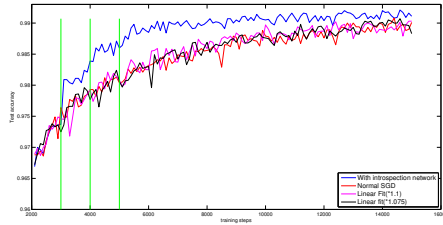

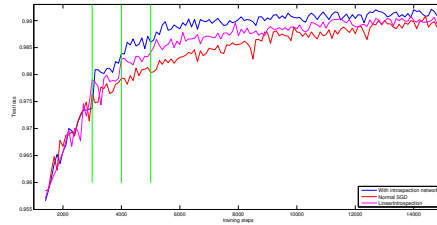

Figure 26: Comparison of test accuracy for $MNIST_1$ with weight updations by Introspection and linear fit.

Figure 27: Validation accuracy plot for $MNIST_1$ using an introspection network without nonlinearity

As no significant improvement in performance was observed the experiment was not repeated over cifar.

### 4.5 LINEAR INTROSPECTION NETWORK

We removed the ReLU nonlinearity from the introspection network and used the same training procedure of the normal introspection network to predict the future values at $2t$. We then used this linear network on the $MNIST_1$ network. We found that it gave some advantage over normal SGD, but was not as good as the introspection network as shown in figure 27. Hence we did not explore this baseline for other datasets and networks.

### 4.5.1 ADDING NOISE

The weight values were updated by adding small gaussian random zero mean noise values . The experiment was performed over $MNIST_1$ for two different std. value, the results of which have been shown below in figure 28.

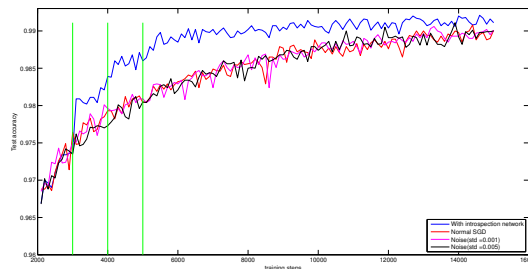

Figure 28: Test accuracy for $MNIST_1$ with weight updations via gaussian noise.

Since no significant improvement was observed for the weight updations via noise for MNIST, the experiment was not performed over cifar-10.

## 5 LIMITATIONS AND OPEN QUESTIONS

Some of the open questions to be investigated relate to determination of the optimal jump points and investigations regarding the generalization capacity of the introspection network to speed up training

in RNNs and non-image tasks. Also, we noticed that applying the jumps in very early training steps while training $AlexNet_1$ tended to degrade the final outcomes. This may be due to the fact that our introspection network is extremely simple and has been trained only on weight evolution data from MNIST. A combination of a more powerful network and training data derived from a diverse set may ameliorate this problem.

## 6   CONCLUSION

We introduced a method to accelerate neural network training. For this purpose, we used a neural network $I$ that learns a general trend in weight evolution of all neural networks. After learning the trend from one neural network training, $I$ is used to update weights of many deep neural nets on 3 different tasks - MNIST, CIFAR-10, and ImageNet, with varying network architectures, activations, optimizers, and normalizing strategies(batch norm,lrn). Using the introspection network $I$ led to faster convergence compared to existing methods in all the cases. Our method has a small memory footprint, is computationally efficient and is usable in practical settings. Our method is different from other existing methods in the aspect that it utilizes the knowledge obtained from weights of one neural network training to accelerate the training of several unseen networks on new tasks. The results reported here indicates the existence of a general underlying pattern in the weight evolution of any neural network.

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

## A  APPENDIX

In this section, we report some initial results of applying the introspection network $I$ (trained on the weight evolution of MNIST network $N0$) to accelerate the training of inception v1 network (Szegedy et al., 2014). We trained the inception v1 network on imagenet dataset with a mini-batchsize of 128 and a RMS optimizer(decay 0.9, momentum 0.9, epsilon 1.0) starting from a learning rate of 0.01 with a decay of 0.94 after every 2 epochs. The network training is still in progress, and we will eventually report on the final outcome. However we thought it would be valuable to share the preliminary results all the same.

We found that applying introspection network seems to be reducing the training time quite significantly. In Figures 29 and 30, we see that applying the introspection network leads to a gain of at least 730,000 steps.After training for around 1.5 million steps, the maximum accuracy achieved by normal training was 68.40%, whereas with introspection applied after every 300k steps the max accuracy achieved was 69.06%.The network achieved the max accuracy of 68.40% after only 852k steps. With introspection applied at steps 200k, 400k and 600k the max accuracy achieved was 68.69% and it reached the max accuracy achieved by the normal training of model after only 944k steps.

However, we also observed that choosing the jump points early in the training does not lead to eventual gains, even though a significant jump in accuracy is observed initially. Figure 31 shows the flattening of the test accuracy after a set of early jumps. It remains to be seen if further interventions later in the training can help maintain the initial accelerated convergence.

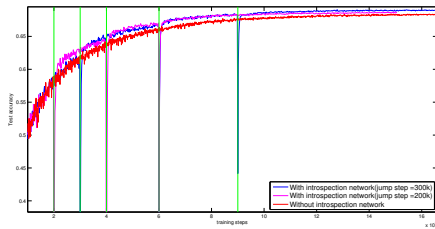

Figure 29: Test accuracy plot for Inception V1 network with weight updates via introspection network at steps $2 \times 10^5$, $4 \times 10^5$ and $6 \times 10^5$(pink curve) and at steps $3 \times 10^5$, $6 \times 10^5$ and $9 \times 10^5$(blue curve)

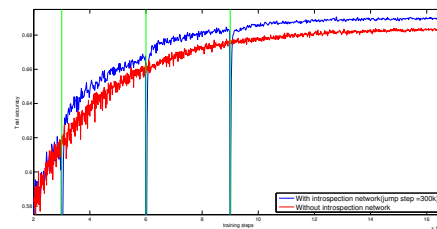

Figure 30: Test accuracy plot for Inception V1 network with weight updates via introspection network at steps $3 \times 10^5$, $6 \times 10^5$ $9 \times 10^5$

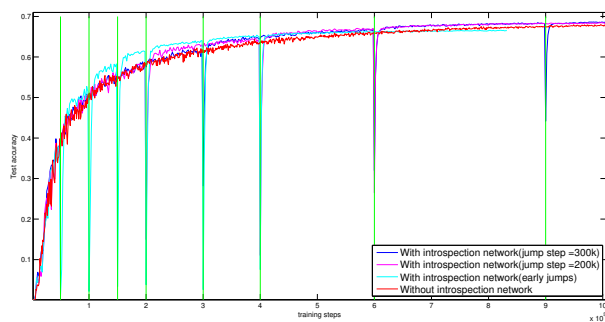

Figure 31: Test accuracy plots for Inception V1 network with weight updates via introspection network in early training steps.

