# Peer review of "Introspection:Accelerating Neural Network Training By Learning Weight Evolution"

_ICLR 2017 — accepted_

[Official Review · AnonReviewer2 · rating 9 · confidence 5 · 17 Dec 2016 (modified: 20 Jan 2017)]
**Valuable insight but needs careful analysis**

EDIT: Updated score. See additional comment.

I quite like the main idea of the paper, which is based on the observation in Sec. 3.0 - that the authors find many predictable patterns in the independent evolution of weights during neural network training. It is very encouraging that a simple neural network can be used to speed up training by directly predicting weights.

However the technical quality of the current paper leaves much to be desired, and I encourage the authors to do more rigorous analysis of the approach. Here are some concrete suggestions:

- The findings in Section 3.0 which motivate the approach, should be clearly presented in the paper. Presently they are stated as anecdotes.

- A central issue with the paper is that the training of the Introspection network I is completely glossed over. How well did the training work, in terms of training, validation/test losses? How well does it need to work in order to be useful for speeding up training? These are important questions for anyone interested in this approach.

- An additional important issue is that of baselines. Would a simple linear/quadratic model also work instead of a neural network? What about a simple heuristic rule to increase/decrease weights? I think it's important to compare to such baselines to understand the complexity of the weight evolution learned by the neural network.

- I do not think that default tensorflow example hyperparameters should be used, as mentioned by authors on OpenReview. There is no scientific basis for using them. Instead, first hyperparameters which produce good results in a reasonable time should be selected as the baseline, and then added the benefit of the introspection network to speed up training (and reaching a similar result) should be shown.

- The authors state in the discussion on OpenReview that they also tried RNNs as the introspection network but it didn't work with small state size. What does "didn't work" mean in this context? Did it underfit? I find it hard to imagine that a large state size would be required for this task. Even if it is, that doesn't rule out evaluation due to memory issues because the RNN can be run on the weights in 'mini-batch' mode. In general, I think other baselines are more important than RNN.

- A question about jump points: 
The I is trained on SGD trajectories. While using I to speed up training at several jump points, if the input weights cross previous jump points, then I gets input data from a weight evolution which is not from SGD (it has been altered by I). This seems problematic but doesn't seem to affect your experiments. I feel that this again highlights the importance of the baselines. Perhaps I is doing something extremely simple that is not affected by this issue.

Since the main idea is very interesting, I will be happy to update my score if the above concerns are addressed.

[Official Review · AnonReviewer3 · rating 8 · confidence 5 · 19 Dec 2016 (modified: 24 Jan 2017)]
**novel idea but requires more details / experimentation**
impact 4 · recommendation (unofficial) 4

The paper reads well and the idea is new.
Sadly, many details needed for replicating the results (such as layer sizes of the CNNs, learning rates) are missing. 
The training of the introspection network could have been described in more detail. 
Also, I think that a model, which is closer to the current state-of-the-art should have been used in the ImageNet experiments. That would have made the results more convincing.
Due to the novelty of the idea, I recommend the paper. I would increase the rating if an updated draft addresses the mentioned issues.

[Official Review · AnonReviewer1 · rating 7 · confidence 4 · 20 Dec 2016 (modified: 23 Jan 2017)]
originality 3 · clarity 5 · meaningful comparison 3

In this paper, the authors use a separate introspection neural network to predict the future value of the weights directly from their past history. The introspection network is trained on the parameter progressions collected from training separate set of meta learning models using a typical optimizer, e.g. SGD.  

Pros:
+ The organization is generally very clear
+ Novel meta-learning approach that is different than the previous learning to learn approach

Cons: 
- The paper will benefit from more thorough experiments on other neural network architectures where the geometry of the parameter space are sufficiently different than CNNs such as fully connected and recurrent neural networks.  
- Neither MNIST nor CIFAR experimental section explained the architectural details
- Mini-batch size for the experiments were not included in the paper
- Comparison with different baseline optimizer such as Adam would be a strong addition or at least explain how the hyper-parameters, such as learning rate and momentum, are chosen for the baseline SGD method. 

Overall, due to the omission of the experimental details in the current revision, it is hard to draw any conclusive insight about the proposed method.

[Author Response · Balaji Krishnamurthy · 06 Feb 2017]
**Updates on Inception V1 Results**

We have revised the paper with updated results on experiments with Inception V1 network, which continues to show promise.

[Final Decision · Program Chairs · 06 Feb 2017]
**ICLR committee final decision**

Interesting paper and clear accept. Not recommended for an oral presentation because of weaknesses in the empirical contribution that make the significance of the results unclear.